

# Effects of moderate physical activity on diabetic adhesive capsulitis: a randomized clinical trial

Raheela Kanwal Sheikh[1,*], Amna Toseef[2,3], Aadil Omer[4,5], Anam Aftab[6], Muhammad Manan Haider Khan[7], Saeed Bin Ayaz[2], Omar Althomli[1], Aisha Razzaq[3], Samra Khokhar[8], Nazia Jabbar[9] and Waqar Ahmed Awan[3,*]

[1] Department of Physiotherapy, College of Applied Medical Sciences, University of Hail, Hail, Saudi Arabia
[2] Physical Medicine & Rehabilitation, Sheikh Khalifa Bin Zayed Al Nahyan Hospital CMH, Muzaffarabad, Azad Kashmir, Pakistan
[3] Faculty of Rehaibilitation & Allied Health Sciences, Riphah International University, Islamabad, islamabad, Pakistan
[4] School of Rehabilitation, Tehran University of Medical Sciences, Tehran, Iran
[5] Islamabad College of Physical Therapy, Margalla Institute of Health Sciences, Islamabad, Pakistan
[6] M. Islam Institute of Rehabilitation Sciences, Gujranwala, Pakistan
[7] Department of Rehabilitation Sciences, Shifa Tamer e Millat University, Islamabad, Pakistan
[8] Nawabshah Institute of Medical and Health Sciences, College of Physical Therapy and Rehabilitation Sciences (NIMHS), Shaheed Benazirabad, Pakistan
[9] Royal Institute of Physiotherapy and Rehabilitation Sciences, Hidayat Campus, Sukkur, Pakistan
* These authors contributed equally to this work.

Corresponding authors
Raheela Kanwal Sheikh,
r.sheikh@uoh.edu.sa
Waqar Ahmed Awan,
waqar.ahmed@riphah.edu.pk

## ABSTRACT

**Background:** Moderate physical activity (MPA) has proven advantages for glycemic control, cardiovascular health, and functional independence. However, physical activity is not part of routine conventional physical therapy (CPT) in managing diabetic adhesive capsulitis patients.

**Objective:** To determine the effects of moderate MPA on diabetic adhesive capsulitis (AC).

**Methodology:** A randomized control trial was conducted at the Combined Military Hospital (CMH), Muzaffarabad, Pakistan from March 2022 to October 2022. A total of $n = 44$ patients with diabetic AC, aged 40 to 65 years, HbA1c > 6.5% were enrolled. Group A received MPA and CPT, while Group B only received CPT for six weeks. The upper extremity function, pain, and range of motion were assessed at baseline, third week, and sixth week through the disability of arm, shoulder, and hand (DASH) questionnaire, numeric pain rating scale (NPRSS), and goniometer respectively.

**Results:** The NPRS score and ROMs showed significant improvement ($p < 0.05$) in group A compared to group B with a large effect size. When comparing the mean difference of the DASH score (73 + 7.21 *vs.* 57.9 + 12.64, $p < 0.001$, Cohen's d = 1.46) was significantly improved with large effect size in group A as compared to group B.

**Conclusion:** MPA along with CPT has positive effects on patient pain, range of motion, and disability in patients with diabetic adhesive capsulitis.

## INTRODUCTION

Adhesive capsulitis (AC) results in pain, stiffness, and a progressive reduction in range of motion in the shoulder joint, which ultimately contributes to a significant decrease in functional active and passive movements (*Le et al., 2017*). A total of 2–5% of the world's population is affected by AC, and women between the ages of 50 and 70 are more likely to get it. The chance of getting AC is increased by several conditions, including thyroid disorders and diabetes mellitus (DM) (*Alhashimi, 2018*; *Eom, Wilson & Bernet, 2022*; *Kim et al., 2023*; *Le et al., 2017*). Patients with DM have a 13.4% prevalence rate for AC and are 20 times more likely to develop it (*Zreik, Malik & Charalambous, 2016*).

Evidence suggested that the individuals with adhesive capsulitis should also be screened for diabetes (*Rai et al., 2019*). Incidence of AC is 2.7% more likely with prolonged poor glycemic control and HbA1c (*Chen et al., 2018*). As inflammatory markers are raised in DM, cause increased growth factor expression and lead to joint synovitis and capsular fibrosis. Raised glucose level enhances glycosylation and the endothelial growth factors slow down the natural inflammatory response of body and makes the disease stay for longer time with worse symptoms (*Sözen et al., 2018*; *Zreik, Malik & Charalambous, 2016*).

Physical therapy techniques including manual therapy exercise therapy and electrotherapy are also strongly recommended for pain relief, improvement of ROM, and functional status (*Chan, Pua & How, 2017*). Physical activity, especially endurance training, is still essential for managing diabetes, as evidenced by improvements in HBA1c, which are linked to inflammation in AC (*Asfaw & Dagne, 2022*; *Struyf, Mertens & Navarro-Ledesma, 2022*; *Zhu et al., 2021*). Aerobics are also found to improve HbA1c, muscle strength, related disease and complication progression (*Williams et al., 2006*). Treadmill use, bicycling or walking 100 min per week at variable speeds are found to have significant effects on diabetics and are preferable (*Qiu et al., 2014*; *Wang et al., 2020*).

Elevated blood glucose levels have the potential to aggravate inflammation and impair the ability to regulate the inflammatory process (*Zreik, Malik & Charalambous, 2016*). The cornerstone of managing diabetes is moderate physical activity, which has proven advantages for glycemic control, cardiovascular health, and general wellbeing (*Asfaw & Dagne, 2022*; *Struyf, Mertens & Navarro-Ledesma, 2022*; *Zhu et al., 2021*). Since the patients with diabetes mellitus are more prone to slow recovery, moderate physical activity lowers the risk of type 2 diabetes and the complications associated with it. At the time of this study, physical activity is not studied or even in practice while managing the adhesive capsulitis in diabetics with physical therapy. Thus, in addition to conventional physical therapy, MPA may help to reduce hyperglycemia and insulin sensitivity which can lead to reduced inflammation and improve adhesive capsulitis. Therefore, the study objective was to determine the effects of added moderate physical activity on diabetic AC. It was

hypothesized that moderate physical activity with conventional physical therapy significantly improves the symptoms of diabetic adhesive capsulitis.

# MATERIALS AND METHODS

## Study design

A randomized clinical trial (NCT04925128) was conducted at the Rehabilitation Department of Sheikh Khalifa Bin Zayed al Nahyan (SKBZ) Combined Military Hospital (CMH), Muzaffarabad, Pakistan from March 2022 to October 2022. The study was approved from the Research Ethical Committee (RIPHAH/RCRS/REC/Letter-01240) of Faculty of Rehabilitation & Allied Health Sciences, Riphah International University. It was carried out after obtaining approval from Ethics Review Committee of SKBZ CMH (Ref. No. Ethical Committee/DME-826). Prior to the study, written informed consent was taken from all participants.

## Participants

A convenient sampling technique was used for sample selection. 100 patients with Adhesive Capsulitis (AC) who visited CMH during the recruitment period were assessed for eligibility. Out of $n = 100$, $n = 51$ patients did not fulfil the criteria and $n = 05$ participants declined to participate due to accessibility issues. So, the $n = 56$ participants were excluded during sampling process. Participants with uncontrolled Type 2 diabetes (HBA1c ≥ 6.5 consistently for more than 6 months) who were on medication for more than 3 years and were physically active both male and female in their daily routine activities but not regularly participating in intense exercises were included. The participants with stage 1 or stage 2 adhesive capsulitis, who ranged in age from 40 to 65, had reduced active and passive range of motion in a capsular pattern, and also had complain of pain (*Pandey & Madi, 2021*) were included in the study. However, the patients with history of shoulder dislocation, lower limb injury, diabetic foot ulcer, diabetic neuropathy, acute or chronic heart disease, rheumatologic disorder, mobility disorder and post-surgical or trauma related patients were excluded during the screening.

## Sample size

A total of $n = 44$ sample size was calculated through G-power, keeping effect size small (0.25) as the physical activity was not studied previously for managing adhesive capsulitis, α error margin at 0.05. To avoid β error probability, the power (1−β) was set at 0.95%. A total of $n = 100$ patients with AC were assessed for eligibility who visited CMH during the recruitment period. The $n = 44$ patients were then randomly divided into group A ($n = 22$) which received moderate physical activity (MPA) on treadmill in addition to conventional physical therapy (CPT) and group B ($n = 22$) received conventional physical therapy (CPT). There were no dropouts during the study (Fig. 1).

## Randomization & blinding

The randomization was done by the individual who was not related to the study. The sealed envelope method was used to allocate the patient randomly. The random numbers
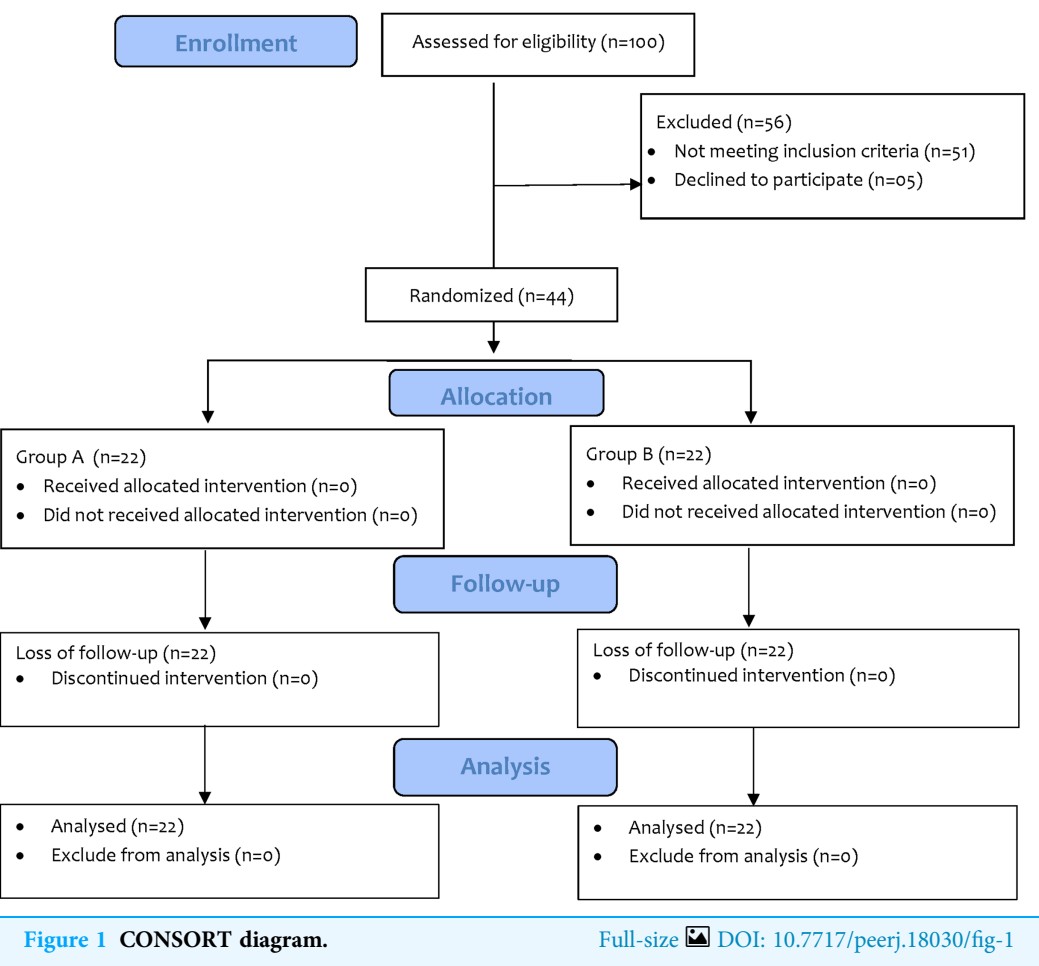

**Figure 1 CONSORT diagram.**               

were generated through the Random Number Generator software. The random numbers were then written on index cards and placed in thick and opaque sealed envelopes before the study. After written informed consent, physical therapist opened the envelope and randomly allocated the patients into respective group. The study was single blinded as the participants were unaware of allocation in the groups (*Malick et al., 2022*).

## Interventions

All patients received a total of 30 sessions, five sessions per week for 6 weeks (*Page & Labbe, 2010*). The duration of session in group A, conventional physical therapy (CPT) and moderate physical activity (MPA), was 60 min, while group B only received CPT had 30 min duration intervention. Before intervention, a brief introduction of interventions was given to both groups. All participants were asked to stop any physical activity during the study period to avoid any confounding effects on the outcomes.

   Group A was instructed to perform physical activity in addition to CPT on treadmill, walking speed and overall handling of the machine. The physical therapist supervised the physical activity on the treadmill to maintain safety and monitor any complication. Initially, walking was started with patient comfortable speed and then increased to the desired level. The participants performed moderate physical activity on treadmill for 5

days in a week for 30 min at 4 mph speed (3–6 METs) with 0% inclination. The warm up and cold down for a 5 min at the beginning and then at the end with speed of 2 mph without inclination (*Umpierre et al., 2011*). The protocol was performed for 6 weeks while maintaining the maximum heart rate between 50–70% of every individual.

Group B received only received CPT including hot pack and transcutaneous electrical nerve stimulation (TENS) for ten minutes. The shoulder was passively mobilized for 10 repetitions in each of the anterior, posterior, and inferior directions while in a pain-free range. The patient actively participated in a variety of stretches, making sure to stay within their pain thresholds. These included shoulder rolls, pendulum stretches, cross-body arm stretches, and towel stretches (10 repetitions in a single session) (*Järvinen et al., 2005*).

### Outcome measures

The average blood glucose level over the previous 6 weeks was measured with haemoglobinA1c (HbA1c) test. The hemoglobin A1c level should fall between 4% and 5.6% in the normal range. Pre-diabetes is identified by a range of 5.7% to 6.4%. When 6.5% or higher, diabetes is present. Validity and reliability have r values of 0.96 and 0.99, respectively (*Carter et al., 1996*).

Shoulder pain was assessed using numeric pain rating scale (NPRS), a reliable (Cronbach's $\alpha = 0.94$) and valid tool (CI [0.96–0.98]) for assessing pain (*Modarresi et al., 2022*).

Shoulder range of motions (ROMs) including abduction, external rotation and internal rotation were measured through the goniometer, has documented reliability of (ICC = 0.92) and validity (r = 0.97, ICC = 0.98) (*Kolber & Hanney, 2012*).

Shoulder functions were evaluated with the Disability of Arm, Shoulder, and Hand (DASH) questionnaire, a reliable and valid tool for upper limb function. Validity and reliability of this scale is (ICC = 0.95) and (ICC = 0.92) respectively (*Kitis et al., 2009*). The data was collected at the baseline, 3$^{rd}$ week and 6$^{th}$ week.

### Statistical analysis

For interaction effects between interventions and assessment level mixed ANOVA was used, while for main effects repeated measure ANOVA was applied with pairwise comparison on NPRS, shoulder range of motion (ROMs) and DASH score. As HBA1c was assessed at baseline and after 6th week, the paired sample t-test was used for within group changes. When comparing the groups on outcome measures, an independent t-test was applied on means of NPRS, ROMs. HBA1c and DASH score were not comparable at the baseline so independent t-test was applied to compare the mean of the mean differences. The partial eta squared, and Cohen's d were used to determine the effect sizes. The $p < 0.05$ was considered as statistically significant value and analyzed by using SPSS version 21.

## RESULTS

Out of $n = 44$ patients the mean age was 52.61 ± 5.81 years. Patients having diabetes for 5.61 ± 3.1 years and mean BMI was 27.2 ± 1.9, $n = 43$ (97.7%) were overweight and $n = 1$ (2.3%) obese. A total of $n = 38$ (86.4%) were females and $n = 6$ (13.6%) were males

**Table 1 Group wise demographic information.**

| | Groups | Mean/n | Std. Deviation/% | p-value |
|---|---|---|---|---|
| Age | Experimental | 51.14 | 5.642 | 0.208 |
| | Control | 53.18 | 4.953 | |
| Gender | | | | |
| Male | Experimental | 4 | 9.09% | 0.31 |
| | Control | 2 | 4.55% | |
| Female | Experimental | 18 | 40.91% | |
| | Control | 20 | 45.45% | |
| Duration of diabetes mellitus (years) | Experimental | 4.500 | 2.4251 | 0.019* |
| | Control | 6.727 | 3.5076 | |
| Body mass index | Experimental | 26.595 | 1.2038 | 0.017* |
| | Control | 27.941 | 2.2466 | |
| Management | | | | |
| Insulin | Experimental | – | – | 0.008** |
| | Control | 4 | 9.09% | |
| Medication | Experimental | 22 | 50% | |
| | Control | 14 | 31.82% | |
| Both | Experimental | – | – | |
| | Control | 4 | 9.09% | |

Note:
Level: $p < 0.05^{*}$, $p < 0.01^{**}$, $p < 0.001^{***}$.

participated in the study. The frequency of insulin dependent patients was $n = 4$ (9.1%), hypoglycemic medication was taken by $n = 36$ (81.8%) and $n = 4$ (9.1%) was on both insulin and hypoglycemic medication. For groupwise distribution please see Table 1.

The results of mixed ANOVA showed significant interaction effect between interventions and assessment level regarding the pain intensity {F = 23.68 (1.75, 73.87), $p < 0.001$, $\eta p^2$ 0.36}, ROM's including abduction ROM {F = 5.94 (1.38, 58.3), $p < 0.001$, $\eta p^2$ 0.12}, External Rotation {F = 8.8 (1.33, 56.1), $p < 0.001$, $\eta p2$ 0.17} and DASH score {F = 22.2 (1.22, 51.3), $p < 0.001$, $\eta p^2$ 0.34}. But non-significant interaction effect regarding internal rotation {F = 1.76 (1.30, 54.8), $p < 0.189$, $\eta p^2$ 0.04} and HbA1c (Fig. 2).

The within-group main effect with RM-ANOVA showed that all variables in group A (MPA+CPT) and B (CPT) were significantly improved from baseline to ($p < 0.001$) six-week and each level of assessment except shoulder internal rotation. HBA1c only improved significantly with a large effect size ($p < 0.001$, Cohen's d = 0.26) in group A, while no significant change ($p \geq 0.05$) was observed in group B (Table 2).

To compare the NPRS Score and ROMs between group A and group B, an independent T-test was used which showed that after, 3rd week and 6th week in pain and all ROMs significant improvement ($p < 0.05$) observed in group A as compared to group B with medium to large effect size. No significant ($p = 0.41$) difference was seen in abduction after 3rd week of intervention (Table 3).

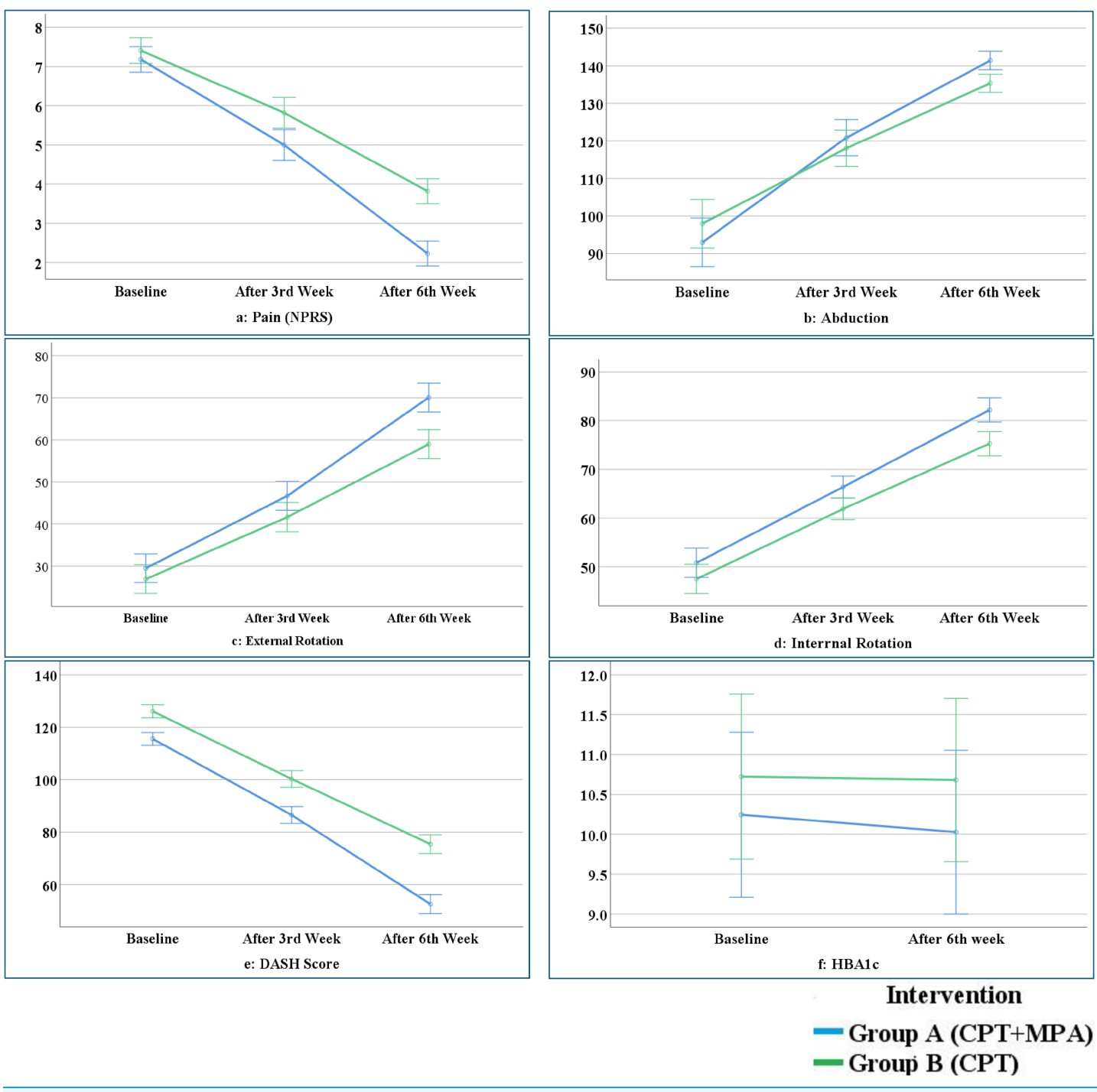

**Figure 2 Interaction effect between variables.**

As HBA1c was not normally distributed and DASH score were not comparable at the baseline, so compared the mean of the mean difference (MD) of HBA1c (0.21 + 0.26 *vs.* 0.04 + 0.14, $p = 0.01$, Cohen's d = 0.21) and DASH score (73 + 7.21 *vs.* 57.9 + 12.64, $p < 0.001$, Cohen's d = 1.46), which showed significant improvement with large effect size in group A as compared to group B (Fig. 3).

**Table 2 With-in group changes.**

| Variable | Assessment | Group A (MPA+CPT) | | | | | Group B (CPT) | | | | |
|---|---|---|---|---|---|---|---|---|---|---|---|
| | | Mean | SD | MD/F(df) | *p*-value | ηp²/ cohen's d | Mean | SD | MD/F(df) | *p*-value | ηp²/ cohen's d |
| NPRS | Baseline | 7.18 | 0.958 | 2.18 | 0.00***a | 0.98 | 7.41 | 0.503 | 1.59 | 0.00***a | 0.91 |
| | 3rd Week | 5.0 | 1.069 | 2.77 | 0.00***b | | 5.82 | 0.733 | 2.0 | 0.00***b | |
| | 6th Week | 2.13 | 0.752 | 1,062 (1.60, 33.5) | 0.00***c | | 3.82 | 0.733 | 232 (1.59, 33.57) | 0.00***c | |
| ROM shoulder abduction | Baseline | 92.95 | 19.18 | −27.9 | 0.00***a | 0.87 | 97.95 | 9.62 | −20.09 | 0.00***a | 0.92 |
| | 3rd Week | 120.8 | 13.57 | −20.5 | 0.00***b | | 118.05 | 8.59 | −17.31 | 0.00***b | |
| | 6th Week | 141.4 | 4.55 | 142.3 (1.36, 28.6) | 0.00***c | | 135.36 | 6.66 | 270.1 (1.47, 30.8) | 0.00***c | |
| ROM shoulder external rotation | Baseline | 29.5 | 8.90 | −17.1 | 0.00***a | 0.97 | 26.91 | 6.83 | −14.7 | 0.00***a | 0.87 |
| | 3rd Week | 46.6 | 8.92 | −23.3 | 0.00***b | | 41.64 | 7.24 | −17.3 | 0.00***b | |
| | 6th Week | 70.5 | 6.47 | 963.2 (1.71, 36.0) | 0.00***c | | 58.95 | 9.36 | 149 (1.24, 26.1) | 0.00***c | |
| ROM shoulder internal rotation | Baseline | 50.82 | 6.85 | −15.4 | 0.00***a | 0.95 | 47.50 | 7.20 | −14.3 | 0.00***a | 0.87 |
| | 3rd Week | 66.36 | 5.61 | −15.8 | 0.00***b | | 61.86 | 4.77 | −13.4 | 0.00***b | |
| | 6th Week | 82.18 | 5.33 | 422 (1.49, 31.4) | 0.00***c | | 75.27 | 6.32 | 147 (1.10, 23.3) | 0.00***c | |
| DASH | Baseline | 79.46 | 6.26 | 31.9 | 0.00***a | 0.988 | 85.68 | 4.95 | 27.3 | 0.00***a | 0.95 |
| | 3rd Week | 47.52 | 5.12 | 28.6 | 0.00***b | | 58.36 | 6.40 | 20.6 | 0.00***b | |
| | 6th Week | 18.82 | 3.05 | 1,687 (1.36, 28.9) | 0.00***c | | 37.74 | 9.07 | 441 (1.15, 24.2) | 0.00***c | |
| HBA1c | Baseline | 10.24 | 2.37 | 0.22 | 0.00*** | 0.26 | 10.72 | 2.47 | 0.04 | 0.18 | 0.14 |
| | 6th Week | 10.02 | 2.32 | | | | 10.68 | 2.47 | | | |

**Notes:**
[a]Baseline week *vs.* week 3rd, [b]week 3rd *vs.* 6th week, [c]baseline week *vs.* 6th week.
NPRS, numeric pain rating scale; Visual Analogue Scale; ROM, range of motion; DASH, disabilities of the arm, shoulder and hand; HBA1c, glycated haemoglobin; CPT, conventional physical therapy; MPA, moderate physical activity.
Differences within groups were analyzed by independent sample t-test.
Significance level-*$p < 0.05$, **$p < 0.01$,***$p < 0.001$.

# DISCUSSION

The objective of study was to determine effects of moderate physical activity (MPA) on diabetic adhesive capsulitis in addition to conventional physical therapy (CPT). The results suggest that participants in both groups showed significant improvement in shoulder pain, range of motions, DASH score. However, significant HbA1c changes were noted only in group A (MPA+CPT). While comparing both groups after 6 weeks, group A showed significant improvement than group B (CPT) in pain, ROMs, overall functionality according DASH score and HBA1c. These participants were almost back to their routines and were able to perform their ADL's and IADL's without any pain and restriction as compared to conventional physical therapy group.

The current study's findings showed that conventional physical therapy (CPT) considerably reduced pain, shoulder range of motions, and upper limb function. According to the literature, conventional physical therapy for adhesive capsulitis attempts to reduce discomfort, expand the range of motion, enhance joint nutrition and lubrication, prevent muscle atrophy, and trigger neurological changes (*Jain & Sharma, 2014*; *Nakandala et al., 2021*). The underlying mechanism of these results include breaking adhesions, stretching constrictive tissues, encouraging joint fluid circulation, enhancing

**Table 3 Between group analysis.**

| Variable | Time Period | Group A (MPA+CPT) | | Group B (CPT) | | MD | *p*-value | Cohens' d |
|---|---|---|---|---|---|---|---|---|
| | | Mean | SD | Mean | SD | | | |
| NPRS | Baseline | 7.18 | 0.958 | 7.41 | 0.503 | −0.227 | 0.330 | – |
| | 3rd Week | 5 | 1.069 | 5.82 | 0.733 | −0.818 | 0.005** | 0.916 |
| | 6th Week | 2.13 | 0.752 | 3.82 | 0.733 | −1.591 | 0.00*** | 0.742 |
| ROM shoulder abduction | Baseline | 92.95 | 19.18 | 97.95 | 9.62 | −5.0 | 0.281 | – |
| | 3rd Week | 120.8 | 13.57 | 118.05 | 8.59 | 2.8 | 0.415 | 11.36 |
| | 6th Week | 141.4 | 4.55 | 135.36 | 6.66 | 6.0 | 0.00*** | 5.7 |
| ROM shoulder external rotation | Baseline | 29.5 | 8.90 | 26.91 | 6.83 | 2.59 | 0.285 | – |
| | 3rd Week | 46.6 | 8.92 | 41.64 | 7.24 | 5.04 | 0.046* | 8.12 |
| | 6th Week | 70.5 | 6.47 | 58.95 | 9.36 | 11.09 | 0.00*** | 8.04 |
| ROM shoulder internal rotation | Baseline | 50.82 | 6.85 | 47.50 | 7.20 | 3.31 | 0.125 | – |
| | 3rd Week | 66.36 | 5.61 | 61.86 | 4.77 | 4.50 | 0.006** | 5.20 |
| | 6th Week | 82.18 | 5.33 | 75.27 | 6.32 | 6.90 | 0.00*** | 5.81 |
| DASH | Baseline | 79.46 | 6.26 | 85.68 | 4.95 | −6.21 | 0.00*** | – |
| | 3rd Week | 47.52 | 5.12 | 58.36 | 6.40 | −10.84 | 0.00*** | – |
| | 6th Week | 18.82 | 3.05 | 37.74 | 9.07 | −18.91 | 0.00*** | – |
| HBA1c | Baseline | 10.24 | 2.37 | 10.72 | 2.47 | 0.22 | 0.51 | – |
| | 6th Week | 10.02 | 2.32 | 10.68 | 2.47 | 0.04 | 0.37 | – |

**Notes:**
NPRS, numeric pain rating scale; Visual Analogue Scale; ROM, range of motion; DASH, disabilities of the arm, shoulder and hand; HBA1c, glycated haemoglobin; CPT, conventional physical therapy; MPA, moderate physical activity.
Differences between groups were analyzed by independent sample t-test.
Significance Level-*p < 0.05, **p < 0.01, ***p < 0.001.

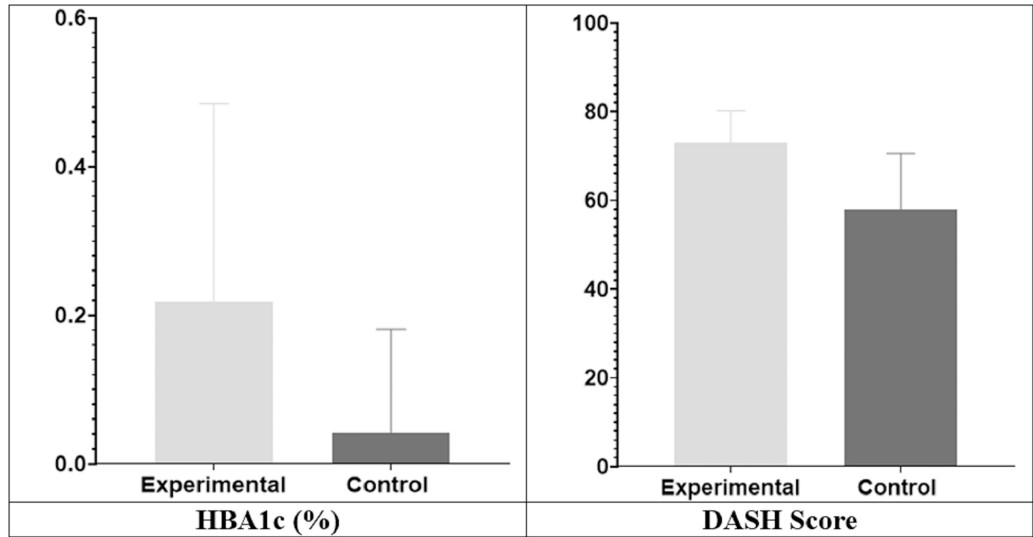

**Figure 3 Comparison of mean difference between groups (HBA1c & DASH).**

muscle strength, and offering the nervous system sensory input (*Page et al., 2014a*, *2014b*; *Shabbir et al., 2021*). The conventional physical therapy for adhesive capsulitis does not have any role in glycemic control because it only focuses on local management of AC. So, in current study the results also depicted the same.

In the current study, the addition of the moderate physical activity along with conventional PT also significantly improved HBA1c as well as the adhesive capsulitis symptoms including pain, ROMs and functional disability. Diabetes may affect the body's ability to regulate blood sugar levels, contributing to the severity of adhesive capsulitis (*Struyf, Mertens & Navarro-Ledesma, 2022*). The accumulation of advanced glycation end-products (AGEs) in the joint tissues due to hyperglycemia may reduce the collagen elasticity as well as microcirculation (*Chen et al., 2018*). Systematic and meta-analysis supported the current study in which effect of moderate physical activity was seen on glycemic control in diabetic patients (*Asfaw & Dagne, 2022*; *Zhu et al., 2021*).

Brisk walking, aerobic training and resistance training had a significant impact on level of HbA1c when conducted for 3 months (*Najafipour et al., 2017*). A systematic review and meta-analysis reviewed the impact on HbA1c in diabetics and it was found that significant differences were seen after at least 12 weeks of intervention (*Bekele et al., 2021*). In the current study HbA1c was compared at the beginning and after 6 weeks, which also showed statistically significant improvement in HBA1c level.

The current study also suggested that in comparison to conventional physical therapy, addition of moderate physical activity showed significantly better results in patients with diabetic adhesive capsulitis patients. MPA may facilitate the improvement in the symptoms by controlling the effects of hyperglycemia and overall metabolic health (*Kanaley et al., 2022*). Better metabolic control can indirectly contribute to improved healing and reduced pain. Physical activity also stimulates the release of endorphins, which are natural pain-relieving chemicals produced by the body and can help reduce pain perception and promote a sense of well-being (*Chen et al., 2022*; *Geneen et al., 2017*).

HbA1c gives an overview of the typical blood sugar levels during the previous three months (*Sherwani et al., 2016*). However, the HbA1c measurement was only performed once after 6 weeks in the study under consideration. Therefore, the shorter research period may make it more difficult to reliably record substantial increases in HbA1c levels. In the current study, patients with diabetic adhesive capsulitis had their shoulder joint's capsular pattern evaluated. The study did not, however, provide any data on the patients' nutritional intake. The management of diabetes and general health can be significantly influenced by dietary considerations (*Rajput, Ashraff & Siddiqui, 2022*). A thorough understanding of the relationship between dietary components and the desired outcomes may be constrained if food intake is not considered in the study.

Limitations: although the results are significant but in current study stage 1 and 2 both were included. Which may affect timeline of improvement in the symptoms at varied rate. Moreover, the history of change in the medication was not observed, which may affect the result significantly.

## CONCLUSIONS

Moderate physical activity (MPA) for the management of diabetic adhesive capsulitis was found to be effective in improving pain and stiffness, range of motion and overall functionality along with reducing HBA1c level. To enhance effectiveness of physical therapy management of diabetic AC, it is recommended that incorporation of moderate physical activity, may improve the clinical effectiveness of physical therapy for such population. Adding dietary recommendations or controlling diet and change in the medication in place together with considering severity level of adhesive capsulitis may provide more valid result on AC symptoms associated with HbA1c levels.

## ACKNOWLEDGEMENTS

Thanks to the participants of this study for sharing their personal experiences with pain.

### Funding

The authors received no funding for this work.

### Competing Interests

The authors declare that they have no competing interests.

### Author Contributions

- Raheela Kanwal Sheikh conceived and designed the experiments, analyzed the data, prepared figures and/or tables, and approved the final draft.
- Amna Toseef conceived and designed the experiments, performed the experiments, authored or reviewed drafts of the article, and approved the final draft.
- Aadil Omer conceived and designed the experiments, prepared figures and/or tables, and approved the final draft.
- Anam Aftab analyzed the data, authored or reviewed drafts of the article, and approved the final draft.
- Muhammad Manan Haider Khan analyzed the data, prepared figures and/or tables, and approved the final draft.
- Saeed Bin Ayaz performed the experiments, authored or reviewed drafts of the article, and approved the final draft.
- Omar Althomli conceived and designed the experiments, authored or reviewed drafts of the article, and approved the final draft.
- Aisha Razzaq analyzed the data, authored or reviewed drafts of the article, and approved the final draft.
- Samra Khokhar performed the experiments, authored or reviewed drafts of the article, and approved the final draft.
- Nazia Jabbar performed the experiments, authored or reviewed drafts of the article, and approved the final draft.

- Waqar Ahmed Awan conceived and designed the experiments, performed the experiments, analyzed the data, prepared figures and/or tables, authored or reviewed drafts of the article, and approved the final draft.

### Human Ethics

The following information was supplied relating to ethical approvals (*i.e.*, approving body and any reference numbers):

The study was approved by the Research Ethical Committee of the Faculty of Rehabilitation & Allied Health Sciences, Riphah International University (RIPHAH/RCRS/REC/Letter-01240).

It was carried out after obtaining approval from the Ethics Review Committee of Sheikh Khalifa Bin Zayed al Nahyan (SKBZ) Combined Military Hospital (CMH), Muzaffarabad, Pakistan (Ref. No. Ethical Committee/DME-826).

### Data Availability

The raw data is available in the Supplemental File.

### Clinical Trial Registration

The following information was supplied regarding Clinical Trial registration:

NCT04925128.

### Supplemental Information

Supplemental information for this article can be found online at http://dx.doi.org/10.7717/peerj.18030#supplemental-information.

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
