# Peer review of "Effects of moderate physical activity on diabetic adhesive capsulitis: a randomized clinical trial"

_PeerJ, doi:10.7717/peerj.18030_

## Round 0.1 · original submission · Major Revisions

Dear Authors:

Thank you for considering PeerJ to submit your paper. After review, we consider "Major Reviews" please attend to all comments.

Regards

Dr. Manuel Jiménez

Reviewer 1 ·

Basic reporting

1. There is a grammatical error in lines no 52-54. Consider revising the statement.
2. The author has mentioned middle age is between 50-70 in line number 55. Provide a reference for this age group or consider revising this statement
3. Provide a reference for the statement that starts….. “The chance of getting” in line no 56
4. Consider to correct the technical error in line no 61.
5. Line no 86….what are the primary objectives? And secondary objectives?
6. Line no 70….” Physical activity”....long statement. Consider revising.
7. Line no 73…. “Aerobics are also”........this statement appears incomplete. Consider revising.
8. Consider to improvise the literature gap and the need for the study. What is the research question?

Experimental design

1. Line no 100….. “Participants”. The total number of patients assessed for eligibility criteria is 100, 56 were excluded. In this case, how was the systematic sampling technique applied?
2. Line no 107….. “Inclusion criteria”.... What was the criteria for stages 1 and 2 of AC were considered? 3. Did the author consider the HBA1C level of the participants as an inclusion criteria? How the physical active level was measured among the participants? What were the criteria for uncontrolled DM considered?
3. Line no 111….. “Sample Size” , justify the proposed sample size.
4. Line no 121….. “Randomization” and “Random allocation” are not explained properly. Consider revising it
5. Line no 131… Group A received only 30 minutes of protocol. Where in Line no 116, it is mentioned that Group A receives physical activity and conventional physiotherapy protocol? According to that they should receive 90 minutes of protocol. Justify.
6. Line no 135 Treadmill protocol appears incomplete. How the intensity of the exercise was considered? 7. The same intensity of the exercises was followed for 6 weeks without any modification. What safety precaution was followed ? Justify

Validity of the findings

1. What are the interpretation criteria for the Cohen’s D effect size considered?
2. Line no 204…. Mean difference interpretation is not reported, consider revising.
3. Line no: 263, Limitation of the study is not appropriate. This study never intended to explore the nutritional aspect. So no need to mention it. Consider revising statement
4. Reframe the conclusion as per the objective of this study.
5. Table No. 1: Replace history with DM to Duration of DM
6. Table No. 3: expand this MPA and CPT term below the table
7. Figure 2: colour coding between the groups is not distinct
8. Figure 3: Error in axis title. Expand the word MD. The picture has to be self-explanatory

Additional comments

1. Overall the manuscript writing has to be improved, pay attention to the grammar.
2. Weak exercise protocol
3. What is the novelty of this study?

·

Basic reporting

no comment

Experimental design

1. Abstract, line 36: intervention for Group A was MPA and CPT and Group B was CPT only but in line 131, the author explains Aerobic Exercise in the Group A intervention. Please explain when the CPT is given in this group.

2. in lines 135-146: group A got aerobic exercise for 30 minutes and Group B got Flexibility Exercise for 60 minutes. Please Explain the reason to compare different types of exercise and the exercise duration of each group.

Validity of the findings

no comment

·

Basic reporting

Review of "Effects of moderate physical activity on diabetic adhesive capsulitis: a randomized clinical trial."

This randomized clinical trial investigated the effect of moderate physical activity (MPA) in addition to conventional physical therapy (CPT) on the pain and stiffness, range of motion, overall functionality and HbA1c level in diabetic patients with adhesive capsulitis. This study showed that MPA was effective for the pain and stiffness, range of motion, overall functionality and HbA1c level in diabetic patients with adhesive capsulitis This study was potentially interesting. This reviewer has several questions and comments.

1. To add the definition of adhesive capsulitis in the Methods section.
2. To add the whether medication was change during the study periods.
3. English needs improvement.

Experimental design

no comment

Validity of the findings

no comment

Additional comments

no comment

---

## Round 0.2 · accepted · Accept

Dear Author:

Your manuscript "Effects of moderate physical Activity on diabetic adhesive Capsulitis: a randomised clinical trial" has been Accepted for publication. However, there is something to change before: please consider this comment from reviewer#2: "The medication affects glycemic control. Thus, there is a possibility that showing HbA1c in the abstract may mislead the general readers".

Congratulations

Dr. Manuel Jiménez

·

Basic reporting

no comment

Experimental design

1. Abstract, line 36: Explain the intervention for Group A (MPA and CPT) and Group B (CPT only) as described in line 131: it's clear
2. Lines 135-146: Explain the reason to compare different types of exercise and the exercise duration of each group : still need to clarify. It's clear

Validity of the findings

no comment

Additional comments

no commment

·

Basic reporting

The medication affect on glycemic control. Thus, there is a possiblity that to show HbA1c in abstract may mislead the general readers.

Experimental design

none

Validity of the findings

none